# Hypoxic Adaptation of Mitochondrial Metabolism in Rat Cerebellum Decreases in Pregnancy

**DOI:** 10.3390/cells9010139

**Published:** 2020-01-07

**Authors:** Anastasia Graf, Lidia Trofimova, Alexander Ksenofontov, Lyudmila Baratova, Victoria Bunik

**Affiliations:** 1Faculty of Biology, Lomonosov Moscow State University, 119234 Moscow, Russia; nastjushka@gmail.com (A.G.); lidtrof@gmail.com (L.T.); 2Faculty of Nano-, Bio-, Informational and Cognitive and Socio-humanistic Sciences and Technologies at Moscow Institute of Physics and Technology, 123098 Moscow, Russia; 3A.N. Belozersky Institute of Physicochemical Biology, Lomonosov Moscow State University, 119992 Moscow, Russia; alexksenofon@gmail.com (A.K.); kukarino@yandex.ru (L.B.); 4Faculty of Bioengineering and Bioinformatics, Lomonosov Moscow State University, 119234 Moscow, Russia; 5Вiochemistry Department, Sechenov University, 119048 Moscow, Russia

**Keywords:** amino acid neurotransmitter, cerebellar amino acid metabolism, hypoxia, 2-oxoglutarate dehydrogenase, tricarboxylic acid cycle

## Abstract

Function of brain amino acids as neurotransmitters or their precursors implies changes in the amino acid levels and/or metabolism in response to physiological and environmental challenges. Modelling such challenges by pregnancy and/or hypoxia, we characterize the amino acid pool in the rat cerebellum, quantifying the levels and correlations of 15 amino acids and activity of 2-oxoglutarate dehydrogenase complex (OGDHC). The parameters are systemic indicators of metabolism because OGDHC limits the flux through mitochondrial TCA cycle, where amino acids are degraded and their precursors synthesized. Compared to non-pregnant state, pregnancy increases the cerebellar content of glutamate and tryptophan, decreasing interdependence between the quantified components of amino acid metabolism. In response to hypoxia, the dependence of cerebellar amino acid pool on OGDHC and the average levels of arginine, glutamate, lysine, methionine, serine, phenylalanine, and tryptophan increase in non-pregnant rats only. This is accompanied by a higher hypoxic resistance of the non-pregnant vs. pregnant rats, pointing to adaptive significance of the hypoxia-induced changes in the cerebellar amino acid metabolism. These adaptive mechanisms are not effective in the pregnancy-changed metabolic network. Thus, the cerebellar amino acid levels and OGDHC activity provide sensitive markers of the physiology-dependent organization of metabolic network and its stress adaptations.

## 1. Introduction

Many amino acids and/or their derivatives are neurotransmitters. Hence, metabolic perturbations in the brain, affecting the levels of amino acids, often have neurological consequences, and vice versa. However, systemic consequences of changed levels of specific amino acids or related enzymes are not easily predictable. For instance, administration of arginine or nitric oxide synthase inhibitors at cerebral infarction may cause opposite physiological outcomes [1]. The glutamate-induced excitotoxicity could be either aggravated [2] or alleviated [1] by nitric oxide signaling. Obviously, one should take into account that generation of nitric oxide involves an intercept between metabolism of lysine and arginine [3], which, in their turn, are tightly linked to other amino acids through multiple intercepts in the amino acid metabolism. In particular, the transporters for the amino acid influx are usually common for a group of amino acids which thus compete for their intracellular transport. 

This work is dedicated to elaboration of systemic markers of the changed metabolism of amino acids, resulting from the brain response to the physiological or pathological challenges. To achieve this goal, we consider the two important features of the amino acid metabolism. First, mitochondrial tricarboxylic acid (TCA) cycle actively participates both in the amino acid degradation and de novo biosynthesis of the amino acid precursors, such as 2-oxoglutarate and oxaloacetate. Under maximal energy demands, the flux through the cycle is limited by the highly regulated multienzyme complex of 2-oxoglutarate dehydrogenase (OGDHC) [4,5,6], which strongly impacts on the amino acid metabolism in the brain and cerebellar neuronal cells in culture [7,8,9,10]. Based on the tight interconnection between the 2-oxo and amino acids, which may contribute to the common neurological symptoms upon the impaired degradation of 2-oxo acids [4,6], we consider dependence of the brain amino acid levels on the OGDHC activity as a systemic marker of mitochondrial metabolism. Second, specific (patho) physiological settings may strongly contribute to different systemic outcomes of the same treatment, because organization of metabolism under these settings may vary. Indeed, the tissue-specific expression of enzymes in a pathway is an important factor in predicting the metabolic changes in health and disease [11], and the expression pattern may vary even in the same tissue in different (patho) physiological states. Indeed, inhibition of 2-oxoglutarate-dehydrogenase, through which glutamate is degraded in the TCA cycle, may increase or decrease the glutamate levels in the rat brain cortex, dependent on pregnancy, which in turn defines the level of OGDHC activity [7,8,12]. Levels of another amino acid of signaling importance, homoarginine, are also affected in pregnancy [13]. Thus, pregnant rats provide a good model of physiological differences in organization of metabolic networks, important for central nervous system functions. On the other hand, influence of specific inhibition of the brain OGDHC on biochemical, physiological, and behavioral parameters of experimental animals strongly depend on the pathological conditions, such as acute hypoxia or ethanol intoxication [12,14]. Because hypoxia is the most common pathogenic factor known to perturb the high-impact signaling by glutamate, (homo) arginine/nitric oxide, and their interaction, we use our well-established model of acute hypobaric hypoxia to study the changes in the brain amino acid metabolism under pathological conditions. 

Because of the significant regional heterogeneity of the brain metabolism and signaling, we focus our investigation on the easily isolated brain structure, cerebellum, which is also well-characterized regarding its physiological functions. In particular, cerebellum is involved in compensatory responses of brain to impaired movement control [15], which also occurs in rats exposed to acute hypobaric hypoxia. The movement disorders in Parkinson disease affecting cerebellum [16,17] have been associated with perturbations in cerebellar pool of amino acids and their signaling [18]. These biochemical changes in cerebellum may be further translated into behavioral changes because of high interconnectivity in the brain [16,19].

In our analysis of systemic response in the healthy and hypoxia-affected brain, we take into account that correlated changes of certain metabolites may provide more information on biosystems than single markers [20,21,22,23]. As a result, the present study demonstrates that the metabolic interdependence of the brain amino acids and OGDHC provides systemic markers of different physiological and pathological states, which complement the information based on analysis of traditional metabolic markers, such as average levels of metabolites or enzymatic activities. Even when single indicators do not significantly change because of homeostatic mechanisms employed by biosystems, the patterns of metabolic correlations reliably estimate systemic changes, helping to suggest the affected pathways.

## 2. Materials and Method

### 2.1. Animal Experiments

All animal experiments were performed according to the Guide for the Care and Use of Laboratory Animals published by the European Union Directives 86/609/EEC and 2010/63/EU, and were approved by Bioethics Committee of Lomonosov Moscow State University (protocol number 69-o from 09.06.2016). Animals were kept at 21 ± 2 °C and relative humidity 53 ± 5% with the 12/12 h light/dark cycle (lights on 9:00 = ZT 0, lights off 21:00 = ZT 12).

Wistar female rats, pregnant and non-pregnant ones, of about 250–300 g were used in the experiment. To obtain pregnant rats, two virgin female rats were located in a cage with one male. After 24 h, vaginal smears were examined. When sperm was found in the vaginal smear, it was considered as the first day of pregnancy, and the male rat was separated. The rats were purchased from the State Research Center of the Russian Federation—Institute for Biomedical Problems, Russian Academy of Sciences, and were kept at our conditions for two weeks prior to the experiments. T/4K cages (555/4K, 580 × 375 × 200 mm) were used. Five to six animals were kept in each cage. Standard rodent pellet food (laboratorkorm.ru) and tap water were available ad libitum.

Four groups of female rats were used: (1) normoxic control non-pregnant group; (2) hypoxic treatment non-pregnant group; (3) normoxic control pregnant group; (4) pregnant group exposed to hypoxia at the 9–10th day of pregnancy, which in rats is roughly correspondent to the first trimester of human pregnancy [24]. Total number of animals in an experimental group, n, comprised animals from several independent experiments and is indicated in the figures and tables. On the second day after the treatment, when the physiological assessment was completed, animals were sacrificed for further biochemical analyses by decapitation. At this stage, the number of pups (10 ± 2) did not significantly differ between the control and hypoxic rats, although independent long-term physiological monitoring indicated that the pregnancy failures occurred more often in the latter than former rats.

Cerebella were quickly excised from animal brains on ice, frozen in liquid nitrogen, and stored at −70 °C prior to biochemical analyses. A half of each thawed cerebellum was used to prepare the homogenates for enzymatic assays, the other half—to make extracts for the amino acid quantification. 

### 2.2. Acute Hypobaric Hypoxia

Female rats were exposed to hypobaric hypoxia at 5% O_2_ (11500 m altitude, 145 mm Hg) in a decompression (altitude) chamber “Mez Mohelnice” (Mohelnice, Czech Republic) of 3.3 L volume, as described previously [25,26]. Briefly, after closing the chamber, air pressure inside the chamber was progressively decreased during 1 min (200 m/sec) by the vacuum pump connected to the chamber. The pressure was controlled using the pressure gauge. Overall response of an organism to hypoxia at subcritical lack of oxygen is defined by the lifetime (LT) which is evaluated from the moment of reaching the target height (i.e., 11,500 m altitude, 5% O_2_, 145 mm Hg) to the second agonal breath. LT characterizes the ability of animals to mobilize their protective mechanisms for survival under life-incompatible extreme conditions. The pressure and oxygen in the chamber returned to the nominal values after the second agonal breath. According to LT, the rats were divided into groups of high resistance (HR, LT ≥ 10 min), middle resistance (MR, LT within 5–10 min), and low resistance (LR, LT within 1–5 min). These types of animals are known to demonstrate different functional and metabolic patterns, including differences in the CNS activity and neurohumoral regulation, stress-responsive systems, oxygen transport by the blood, and tissue respiration [27,28,29].

### 2.3. Enzyme Assays

Homogenization of cerebella and assays of the overall OGDHC activity of brain homogenates were performed as described earlier [30] with 20% glycerol added to the homogenization buffer and sonication as in [12].

### 2.4. Ninhydrine Quantification of Amino Acids

Preparation of the acetic acid/methanol extracts of cerebella and quantification of their amino acids was done as described in [31], using L-8800 amino acid analyzer (Hitachi, Tokyo, Japan) in the standard mode according to the manufacturer′s User Manual (Hitachi High-Technologies Corporation, Tokyo, Japan, 1998). The samples were stored at −70 °C. Briefly, the extracts were subjected to an ion-exchange column 2622SC (PH) (Hitachi, Ltd., Tokyo, Japan, P/N 855-3508, 4.6 × 80 mm), eluted by step gradient of four sodium-acetate buffers at a flow rate of 0.4 mL/min at 57 °C. A total of 20 μL of a 25-fold diluted amino acid standard mix (AA-S-18 −5 ML analytical standard, SIGMA, or standard of basic amino acids, Type B, Hitachi, 016-08641) or 50 μL of cerebellar extracts were injected to the column. Fifteen amino acids eluted as separate peaks in this procedure were quantified. Post-column derivatization (136 °C, flow rate 0.35 mL/min) was performed using the mix of equal volumes of ninhydrin buffer R2 and ninhydrin solution R1 (Wako Pure Chemical Industries, P/N 298-69601). Colored products were detected by absorption at 570 nm. Data were processed using MultiChrom for Windows software (Ampersand Ltd., Moscow, Russia). 

### 2.5. Data Acquisition and Statistics

Statistical analysis was performed using Statistica 10.0 (StatSoft Inc., Tulsa, OK, USA). Averaged values are presented as means ± SEM. Comparison between the two experimental groups was done using non-parametric Mann–Whitney U-test. The Pearson′s correlations between the levels of different amino acids or between the levels of amino acids and OGDHC activity were characterized by the correlation coefficients and *p*-values of the correlation. Statistical significance of differences in the parameters characterizing metabolic interactions between the levels of OGDHC activity and/or amino acids were assessed by the Wilcoxon signed rank test. Differences with *p* ≤ 0.05 were considered significant.

## 3. Results

### 3.1. Pregnancy-Induced Changes in Cerebellar Levels of Amino Acids and OGDHC Activity

Table 1 compares the average levels of the OGDHC activity and 15 quantified amino acids in cerebella of the non-pregnant and pregnant rats. 

The data presented in Table 1 show that most of the differences between the amino acid pools in cerebella of the non-pregnant and pregnant rats are within the quantification errors. However, pregnancy increases the levels of cerebellar glutamate and tryptophan by 22 and 57%, respectively. 

### 3.2. Hypoxia-Induced Changes in Cerebellar Pool of Amino Acids Depend on Physiological State

Table 2 shows that hypoxia induces significant changes in the cerebellar pool of amino acids of non-pregnant female rats. Half of the quantified amino acids (Arg, Glu. Lys, Met, Phe, Ser, Trp) undergo statistically significant increases. 

In contrast, under the same metabolic stress of pregnant rats, their cerebellar levels of amino acids remain unaffected (Table 2). Remarkably, the tryptophan level, which is increased by pregnancy, exhibits a trend (0.08) to decrease (from 0.11 to 0.09 µmol per g FW) after exposure of the pregnant rats to hypoxia, while in the non-pregnant rats hypoxia induces an opposite change in the tryptophan level, which increases from 0.07 to 0.12 µmol per g FW (Table 2). 

### 3.3. Interdependence of Cerebellar Levels of OGDHC Activity and/or Amino Acids Varies under Different Physiological Settings

Metabolic pathways of amino acids have not only common transporters for certain groups of amino acids, but also many common substrates. For instance, transaminase reactions link the levels of the corresponding pairs of the amino acids (aspartate and glutamate in the aspartate transaminase reaction), GABA is produced from glutamate etc. As noted in the introduction section, the amino acid metabolism in general is tightly coupled to the TCA cycle, whose metabolic flux is limited by the activity of OGDHC [4]. As a result, OGDHC inhibition strongly affects the amino acid pool in cerebellar granule neurons in culture [9] and amino acids levels in animal brain in vivo (reviewed in [6]). These metabolic features should cause interdependence of the tissue levels of amino acids. The interdependence is manifested in the correlated variations in the levels of amino acids and OGDHC activity in different animals, because the interindividual variability in the biochemical parameters is contributed by minor variations in the metabolic network organization. Indeed, such correlation analysis of the levels of the cerebellar OGDHC activity and 15 quantified amino acids in a sample of the non-pregnant and pregnant (Table 3) rats reveals a number of highly significant correlations between the studied biochemical parameters, pointing to a strongly interdependent content of different amino acids in rat cerebellum, imposed by intercepts in their metabolism.

Visual inspection indicates that the correlation patterns change along with physiological settings (Table 3, the lower left triangles in A and B). The sets of strongly interacting parameters differ between the non-pregnant and pregnant rats. For instance, pregnancy drastically decreases the interdependences of Glu, Phe, Met with other amino acids and OGDHC, compared to those manifested in the cerebellum of non-pregnant rats. On the other hand, some features of the metabolic network are preserved in both the non-pregnant and pregnant rats. That is, lysine and tryptophan show a low number of interactions in both physiological states (Table 3). Independent of pregnancy, none of the cerebellar amino acids show statistically significant correlation with OGDHC activity, although cerebellar tryptophan tends to positively correlate with OGDHC in both non-pregnant (R = 0.79, 0.061) and pregnant (R = 0.85, 0.07) rats (Table 3). 

In our previous work [23], an approach to quantify subtle physiological shifts in metabolic networks, based on the data of correlation analysis, has been developed. It employs reduction of the correlation matrix dimension, resulting in a small number of easily comparable parameters presented in Table 4. The parameters are: the summarized and averaged correlation coefficients for each element of the matrix, which characterize an overall degree of interdependence of the elements in a specific (patho) physiological state, and the total number of significant (*p* < 0.05) correlations (positive or negative) in the matrix, which indicates how many strong and very strong interdependences of the studied elements are inherent in a metabolic network. According to the data presented in Table 4, the summarized and averaged correlation coefficients are significantly lower in the pregnant rats, compared to the non-pregnant ones, pointing to generally lower interdependences in the amino acid levels in the former than the latter. The lower interdependence is consistent with the lower number of statistically significant correlations in the pregnant vs. non-pregnant rats (38 vs. 62 for the positive correlations and 0 vs. 4 for the negative correlations, Table 4), although this difference does not reach statistical significance.

### 3.4. Concerted Hypoxia-Induced Shift to the Negative Correlations between the Levels of Amino Acids and OGDHC Activity Is not Observed in Pregnancy

Exposure of rats to an environmental challenge, such as acute hypobaric hypoxia, strongly changes the correlation matrices (Table 3, the upper right triangles) and their overall parameters (Table 4). Moreover, the hypoxia-induced changes in the interdependences are well-detectable in both the non-pregnant and pregnant rats (Table 3 and Table 4), in contrast to the changes in the average levels of amino acids and OGDHC activity, which are detectable in the non-pregnant rats only (Table 2). That said, the correlation analysis reveals a very different response to hypoxia of the interdependences between OGDHC activity and/or amino acids in the non-pregnant and pregnant rats. Thus, not only the control correlation matrices, but also those after exposure of animals to hypoxia are very different in non-pregnant and pregnant rats (Table 3, the upper right triangles). As shown in Table 3 hypoxia induces a high number of negative correlations of amino acids with OGDHC in non-pregnant rats. Induction by hypoxia of the negative correlations between the amino acid levels and OGDHC activity exclusively in non-pregnant rats (Table 3, the upper right triangle) thus coincides with the hypoxic reactivity of the average levels of amino acids in non-pregnant rats only (Table 2). In contrast, in pregnant rats hypoxia induces one positive correlation between OGDHC and methionine and one negative correlation between the amino acids tryptophan and phenylalanine (Table 3, the upper right triangle). 

The overall interdependence parameters (Table 4) point to statistically significant differences in the hypoxic responses of cerebellum in both the non-pregnant and pregnant rats. This finding indicates that analysis of the correlation matrices (Table 4) detects the responses to hypoxic stress with a much higher sensitivity, compared to the average levels of amino acids, which did not reveal any changes in the cerebellum of pregnant rats after hypoxia (Table 2). Yet both markers of hypoxic stress point to different responses to hypoxia in the different physiological states. The strong negative correlations between the levels of amino acids and OGDHC activity after hypoxic exposure of non-pregnant rats (Table 3) coincide with the hypoxia-increased levels of the amino acids in these rats (Table 2) and with the overall increase in the interdependence between the OGDHC activity and/or amino acid levels, indicted by the summarized and averaged correlation coefficients and number of statistically significant correlations (Table 4). Contrary, in pregnant rats, demonstrating no changes in the average levels of amino acids after hypoxia (Table 2), no negative correlations of the amino acid levels with OGDHC activity are induced by hypoxia (Table 3), and the overall interdependence is diminished, based on the summarized and averaged correlation coefficients between the studied parameters (Table 4). Remarkably, the number of statistically significant positive correlations between the parameters is increased by hypoxia in pregnant rats too (from 38 to 63, Table 4). However, in accordance with a generally decreased interdependence, evident from the decreased correlation coefficients (summarized and average, Table 4), the increased number of statistically significant positive correlations is expressed less than in non-pregnant rats (Table 4). The most significant contribution to increases in the positive correlations is provided by different amino acids; in non-pregnant rats they are histidine, lysine, and tyrosine, whereas in pregnant rats they are methionine, serine, and phenylalanine (Table 3 and Table 4). Thus, hypoxia affects different metabolic pathways in the cerebella of pregnant and non-pregnant rats.

### 3.5. Physiological Consequences of the Hypoxia-Induced Changes in the Interdependent Levels of OGDHC Activity and/or Amino Acids in Cerebellum

In hypoxic experiments, rats are exposed to acute hypobaric hypoxia until they collapse, which is registered as apnea. As described in the methods section, the time between the established hypoxic condition (5% O_2_) and apnea, i.e., the life time (LT), characterizes individual and/or group differences in the resistance to hypoxia according to the arbitrary intervals of the LT, indicated in the legend to Figure 1. Figure 1 shows that distribution of a sample of the rats into the animals with varied resistance to hypoxia differs dependent on the physiological state. Pregnant rats are characterized by decreased fraction of the animals with high resistance to hypoxia, with the corresponding increase in the fraction of the low resistant rats. The difference is also manifested in a higher average value of LT for non-pregnant rats (369 ± 45.1 s), compared to LT of the pregnant rats (277 ± 37 s) (n = 41; 0.048, according to Mann-Whitney test). Thus, the reactivity to hypoxia is higher (Figure 1) when no significant changes in the OGDHC activity or amino acid levels, neither any negative correlations between these parameters occur in the cerebellum, as observed in the pregnant rats (Table 2, Table 3 and Table 4).

Accordingly, the hypoxia-induced changes in the average levels (Table 2) and/or interdependence (Table 3 and Table 4) of cerebellar OGDHC activity and amino acids, which are observed in the non-pregnant rats, are of compensatory significance. That is, the changes in the amino acid pool (Table 2) and its dependence on OGDHC activity (Table 3), observed in the non-pregnant rats and absent in the pregnant ones, obviously enable the former to resist hypoxia better than the latter (Figure 1).

## 4. Discussion

In this work, cerebellar metabolism of amino acids and its dependence on the TCA-cycle-limiting OGDHC are shown to be affected by physiological settings (pregnancy) and metabolic stress (acute hypobaric hypoxia). Pregnancy increases cerebellar levels of glutamate and tryptophan (Table 1) and decreases overall interdependence of the studied components of amino acid metabolism, compared to their interdependence in non-pregnant rats (Table 4).

Hormonal changes during pregnancy are known to coordinate a broad range of physiological adaptations, from the supply of nutrients and oxygen for the fetus growth in utero to specific patterns of parental behavior [32,33]. Changed metabolism of amino acids in the brain of the pregnant females may be involved in these adaptations, as many of amino acids are neurotransmitters or their precursors. The increases in cerebellar glutamate and tryptophan in pregnant rats, observed in this work, are consistent with independent studies on the pregnancy-imposed changes in glutamatergic and serotonergic signaling. In fact, pregnancy is known to change expression of glutamate receptors to address specific physiological challenges faced by pregnant females [34,35]. Our data on the pregnancy-increased cerebellar content of tryptophan are in line with the activation of serotonergic signaling observed during pregnancy, because tryptophan is the serotonin precursor [36]. The interaction between the levels of serotonin and estrogens also underlies sex-dimorphic prevalence of the serotonin-linked diseases, including migraine, depression, eating disorders and pregnancy-associated pathologies [36,37]. 

Our previous studies pointed to certain relationship between the amino acid levels in the blood plasma and brain [23], which may be used for translation of findings on the brain samples in animal models to human studies. For instance, increased content of tryptophan in cerebellum of pregnant rats, shown in our study (Table 1) corresponds to the findings in humans, which indicate that in maternal plasma, tryptophan catabolites and related compounds change in pregnancy [38,39]. This study suggests the neurotransmitters-dependent adaptation of maternal body to the fetus growth. As considered above, such adaptation is obvious also from our findings (Table 1). Biomarker significance of the amino acid levels in human plasma and urine has also been explored in other studies [40,41,42]. Their findings point to the potential diagnostic significance of the changes in multivariate metabolic profiles, including amino acids, for prediction of gestational diabetes [42]. Besides, the pregnancy-dependent changes in the content of amino acids and their metabolites in plasma and urine point to variation of these parameters, dependent on the increased amino acid demands for the growing fetus [40,41].

The pregnancy-induced changes in cerebellar pool of amino acids (Table 1) are associated with different responses of cerebellar amino acid metabolism to hypoxia in the pregnant and non-pregnant rats (Table 2, Table 3 and Table 4). Strong negative correlations between the amino acid levels and OGDHC activity, which are a hallmark of cerebellar metabolism after hypoxia of non-pregnant rats, are not induced by hypoxia in the pregnant rats (Table 3 and Table 4). Simultaneously, hypoxia significantly increases average levels of cerebellar amino acids in the non-pregnant rats only (Table 2). Because the observed biochemical changes in the cerebellar metabolism of amino acids are associated with a higher resistance to hypoxia in the non-pregnant vs. pregnant rats (Figure 1), the metabolic rearrangement in the cerebellum of non-pregnant rats is of adaptive significance. In particular, increased flux of the amino acids degraded through OGDHC (Glu, Gln, Arg, His, Pro) and of the branched-chain amino acids Val and Ile may generate succinyl-CoA for the substrate level phosphorylation in mitochondria. Generation of ATP at the expense of succinyl-CoA may help overcoming energy deficits upon hypoxia which impairs oxidative phosphorylation. Increased degradation of amino acids in the TCA cycle whose flux is limited by OGDHC, is in good accordance with the negative correlations between the OGDHC activity and the levels of cerebellar amino acids after hypoxia, in contrast to normal metabolism (Table 3). It is worth noting in this regard that hypoxic tolerance is associated with the mTOR-dependent autophagy [43]. Increased autophagy may generate the higher amino acid levels after hypoxia (Table 2) to use them for the substrate-level ADP phosphorylation in the hypoxic brain of non-pregnant rats. Autophagy is also coupled to pro-survival function of mitochondrial fission under energy stress [44,45,46,47]. However, the homeostatic and pro-survival functions of mTOR, autophagy and mitochondrial fission are highly conditional, with overactivation of these processes also mediating the brain damage by hypoxia [48,49,50]. 

Different levels of activation of mTOR, autophagy and mitochondrial fission may be required for their pro-survival and death-inducing outcomes in different physiological states. In this regard, no increase in the brain amino acid levels by hypoxic exposure of pregnant rats (Table 2) was due to a higher damaging potential of the autophagy stimulation in this physiological state. No adaptation to hypoxia by increased degradation of amino acids through OGDHC in pregnant rats is also evident from the hypoxia-induced decrease in the interdependences between the levels of OGDHC activity and/or amino acids, whereas in the non-pregnant rats hypoxia increases these interdependences (Table 4). The different action of hypoxia in the two physiological states is obvious from comparison of the hypoxia-induced changes in the summarized and average correlation coefficients or in the number of significant correlations in the pregnant and non-pregnant rats (Table 4). In view of the increased sensitivity of pregnant vs. non-pregnant rats to hypoxia (Figure 1), the stability of the biochemical parameters in the cerebellum of the pregnant rats exposed to hypoxia (Table 2) along with the absence of negative correlations between the OGDHC activity and levels of amino acids (Table 3 and Table 4) manifest limitations of the hypoxic adaptation through increased degradation of amino acids in the pregnant vs. non-pregnant rats. These findings indicate that the pregnancy-imposed changes in the cerebellar amino acid pool (Table 2) and metabolism (Table 3 and Table 4) are associated with decreased stress adaptability, in line with other studies showing decreased perception of stress in pregnant females [51,52,53]. 

It is worth noting that the response of the metabolic network of pregnant rats to hypoxia, undetectable from the average levels of amino acids (Table 2), is evident from the correlation analysis (Table 4). Hence, average levels of cerebellar amino acids are not as sensitive indicators of metabolic changes, as the cumulative parameters characterizing the interdependence of components of the corresponding metabolic network. 

It has been noted previously that correlating metabolites may not only be of diagnostic significance [23], but also help deciphering the yet unknown or poorly characterized synthetic and regulatory pathways [20]. In this regard, the pregnancy-induced changes in metabolic correlations of cerebellar lysine are of interest, because gestational diabetes is associated with plasma levels of lysine and tyrosine [54], both of them correlating to each other much stronger in cerebellum of the pregnant than non-pregnant rats (Table 3). Besides, lysine catabolism is related to biosynthesis of homoarginine which is elevated in normal pregnancy [13], with some studies linking its elevation to pregnancy disorders, including preeclampsia [55]. Physiological manifestations of pre-eclampsia and eclampsia, associated with nearly one-tenth of all maternal deaths [56], involve changed cerebral hemodynamics and hypertensive encephalopathy [57,58,59], potentially linked to impaired signaling by homoarginine, known as a predictor of cardiovascular risk and mortality [3]. Alternatively, homoarginine is synthesized from arginine and glycine. Because these amino acids are highly correlated independent of pregnancy and/or hypoxia (Table 3), metabolism of cerebellar lysine appears to be a more likely contributor to specific adaptations imposed by pregnancy. This is further supported by the fact that lysine is known as an antagonist of a serotonin receptor [60], with serotonergic signaling increased in pregnancy [36,37] and depressed upon increased synthesis of homoarginine from lysine [61].

## 5. Conclusions

Markers of systemic changes in the cerebellar metabolism of amino acids were introduced, showing decreased reactivity of the pregnancy-changed amino acids network to metabolic stress, such as acute hypobaric hypoxia.

## Figures and Tables

**Figure 1 cells-09-00139-f001:**
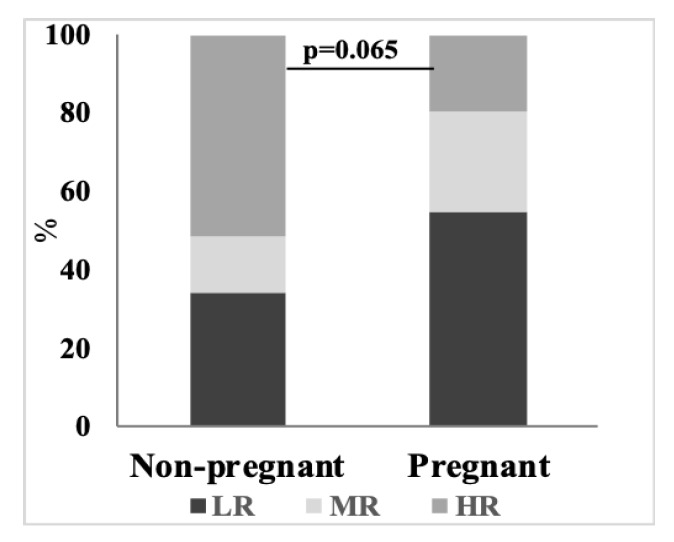
Distribution of the non-pregnant (*n* = 41) and pregnant (*n* =31) rats according to their relative resistance to acute hypobaric hypoxia. HR—highly resistant rats (dark-grey), spent under hypoxia ≥10 min; MR—medium resistant rats (light-grey), spent under hypoxia between 5 and 10 min; LR—low resistant rats (middle-grey), spent under hypoxia ≤5 min. % of each sub-group to total number of animals in the group is shown in shades of grey. The indicated *p* value is estimated using Fisher exact test.

**Table 1 cells-09-00139-t001:** Influence of pregnancy on the levels of cerebellar amino acids and 2-oxoglutarate dehydrogenase complex (OGDHC) activity.

Parameter	Rats		
	Non-Pregnant	Pregnant	*p* Values
**OGDHC**	1.13 ± 0.13	1.21 ± 0.15	0.61
**ALA**	0.59 ± 0.05	0.62 ± 0.04	0.67
**ARG**	0.15 ± 0.03	0.15 ± 0.02	0.22
**ASP**	1.96 ± 0.09	2.01 ± 0.11	0.32
**GABA**	1.72 ± 0.18	1.63 ± 0.16	0.67
**GLU**	***9.11 ± 0.36***	***11.00 ± 0.39***	***0.02***↑
**GLY**	0.82 ± 0.10	0.75 ± 0.10	0.47
**HIS**	0.06 ± 0.01	0.07 ± 0.01	0.19
**ILE**	0.03 ± 0.01	0.03 ± 0.01	1.00
**LEU**	0.08± 0.02	0.07 ± 0.01	0.71
**LYS**	0.35 ± 0.02	0.37 ± 0.02	0.24
**MET**	0.05 ± 0.00	0.06 ± 0.01	0.59
**PHE**	0.06 ± 0.01	0.08 ± 0.02	0.16
**SER**	0.61 ± 0.03	0.56 ± 0.03	0.24
***TRP***	***0.07 ± 0.00***	***0.11 ± 0.03***	***0.02***↑
**TYR**	0.07 ± 0.01	0.07 ± 0.01	0.89

Amino acids are given in alphabetical order. Proteinogenic amino acids are indicated by their standard abbreviations, GABA-γ-aminobutyric acid. Levels of the amino acids and OGDHC activity are presented as mean ± SEM in μmol and μmol per min, correspondingly, per g of the tissue fresh weight. *p* values indicate significance of the differences between the two groups, estimated by the Mann-Whitney U-test. Significant (*p* ≤ 0.05) differences are indicated in bold italics. Seventeen non-pregnant and eight pregnant rats were used in the comparison.

**Table 2 cells-09-00139-t002:** Influence of acute hypobaric hypoxia on the OGDHC activity and amino acid pool in cerebella of the non-pregnant and pregnant rats.

	Groups	Non-Pregnant Rats			Pregnant Rats		
Parameters		Control	Hypoxia	*p* Values	Control	Hypoxia	*p* Values
**OGDH**		1.13 ± 0.13	1.11 ± 0.18	0.561	1.21 ± 0.15	1.24 ± 0.18	0.875
**ALA**		0.59 ± 0.05	0.73 ± 0.06	0.081	0.62 ± 0.04	0.60 ± 0.04	0.633
**ARG**		***0.15 ± 0.03***	***0.23 ± 0.04***	***0.048***↑	0.15 ± 0.02	0.16 ± 0.02	0.762
**ASP**		1.96 ± 0.09	2.30 ± 0.19	0.180	2.01 ± 0.11	2.08 ± 0.09	0.573
**GABA**		1.72 ± 0.18	2.22 ± 0.25	0.091	1.63 ± 0.16	1.54 ± 0.11	0.460
**Glu**		***9.11 ± 0.36***	***10.93 ±0.61***	***0.027***↑	11.00 ± 0.01	10.56 ± 0.42	0.274
**GLY**		0.82 ± 0.10	1.03 ± 0.14	0.180	0.75 ± 0.10	0.77 ± 0.08	1.000
**HIS**		0.06 ± 0.01	0.08 ± 0.01	0.055	0.07 ± 0.01	0.07 ± 0.01	0.696
**ILE**		0.03 ± 0.01	0.05 ± 0.01	0.162	0.03 ± 0.01	0.04 ± 0.01	0.696
**LEU**		0.08± 0.02	0.12 ±0.02	0.116	0.07 ±0.01	0.10 ± 0.02	0.829
**LYS**		***0.35 ± 0.02***	***0.42 ± 0.03***	***0.048***↑	0.37 ± 0.02	0.39 ± 0.03	0.515
**MET**		***0.05 ± 0.00***	***0.09 ± 0.00***	***0.014***↑	0.06 ± 0.01	0.07 ± 0.01	0.573
**PHE**		***0.06 ± 0.01***	***0.13 ± 0.02***	***0.004***↑	0.08 ± 0.02	0.09 ± 0.01	0.829
**SER**		***0.51 ± 0.03***	***0.61 ± 0.02***	***0.036***↑	0.56 ± 0.03	0.56 ± 0.05	0.274
**TRP**		***0.07 ± 0.00***	***0.10 ± 0.01***	***0.048***↑	0.11 ± 0.03	0.09 ± 0.01	0.083
**TYR**		0.07 ± 0.01	0.08 ± 0.01	0.351	0.07 ± 0.01	0.07 ± 0.01	1.000

Amino acids are abbreviated as in Table 1. Levels of amino acids and OGDHC activity are presented as mean ± SEM in μmol and μmol per min, correspondingly, per g of the tissue fresh weight. *p* values indicate significance of the differences between the two groups, estimated by the Mann-Whitney U-test. Significant (*p* ≤ 0.05) differences are shown in bold italics. Number of animals in the groups: 17 control non-pregnant, 8 hypoxic non-pregnant, 8 control pregnant and 10 hypoxic pregnant.

**Table cells-09-00139-t003a:** 

(A) Non-Pregnant Rats																	
	H	OGDHC	ALA	ARG	ASP	GABA	GLU	GLY	HIS	ILE	LEU	LYS	MET	PHE	SER	TRP	TYR
C																	
**OGDHC**			***−0.94*** 0.001	***−0.94*** 0.001	***−0.90*** 0.002	***−0.95*** 0.000	***−0.84*** 0.009	***−0.95*** 0.000	***−0.94*** 0.000	***−0.95*** 0.000	***−0.95*** 0.000	−0.69 0.059	−0.65 0.084	***−0.95*** 0.000	−0.19 0.649	0.53 0.181	***−0.89*** 0.003
**ALA**		−0.42 0.407		**0.99** 0.000	**0.86** 0.006	**0.97** 0.000	**0.82** 0.013	**0.96** 0.000	**0.94** 0.000	**0.97** 0.000	**0.98** 0.000	**0.81** 0.015	**0.71** 0.046	**0.99** 0.000	0.31 0.452	−0.68 0.065	**0.93** 0.001
**ARG**		−0.62 0.187	**0.87** 0.025		**0.85** 0.008	**0.98** 0.000	**0.83** 0.010	**0.99** 0.000	**0.94** 0.000	**0.98** 0.000	**0.98** 0.000	**0.78** 0.022	0.62 0.103	**0.97** 0.000	0.27 0.518	−0.66 0.078	**0.97** 0.000
**ASP**		−0.68 0.135	0.78 0.068	**0.97** 0.001		**0.91** 0.001	**0.94** 0.001	**0.87** 0.005	**0.91** 0.002	**0.89** 0.003	**0.88** 0.004	**0.80** 0.017	**0.71** 0.050	**0.86** 0.007	0.56 0.147	−0.33 0.422	**0.86** 0.006
**GABA**		−0.63 0.178	**0.91** 0.011	**0.92** 0.009	**0.93** 0.008		**0.92** 0.001	**0.99** 0.000	**0.94** 0.000	**0.97** 0.000	**0.96** 0.000	**0.80** 0.017	0.60 0.116	**0.97** 0.000	0.35 0.397	−0.56 0.152	**0.97** 0.000
**GLU**		−0.42 0.402	0.76 0.081	**0.84** 0.036	**0.90** 0.014	**0.90** 0.014		**0.89** 0.003	**0.90** 0.003	**0.85** 0.007	**0.84** 0.009	**0.84** 0.009	0.46 0.246	**0.83** 0.011	0.53 0.176	−0.39 0.340	**0.90** 0.002
**GLY**		−0.64 0.170	**0.86** 0.027	**0.98** 0.000	**0.96** 0.002	**0.94** 0.005	**0.83** 0.040		**0.95** 0.000	**0.98** 0.000	**0.97** 0.000	**0.76** 0.030	0.53 0.172	**0.97** 0.000	0.25 0.546	−0.56 0.148	**0.96** 0.000
**HIS**		0.03 0.955	−0.05 0.918	0.12 0.814	0.06 0.911	−0.05 0.932	−0.17 0.750	0.22 0.671		**0.97** 0.000	**0.98** 0.000	**0.85** 0.008	0.64 0.085	**0.93** 0.001	0.39 0.340	−0.58 0.132	**0.93** 0.001
**ILE**		−0.72 0.106	**0.93** 0.008	**0.91** 0.011	**0.87** 0.025	**0.95** 0.003	0.74 0.093	**0.93** 0.007	0.02 0.970		**0.99** 0.000	**0.78** 0.024	0.64 0.091	**0.97** 0.000	0.30 0.468	−0.56 0.151	**0.94** 0.000
**LEU**		−0.70 0.125	**0.94** 0.005	**0.90** 0.015	**0.86** 0.028	**0.96** 0.002	0.76 0.078	**0.91** 0.011	−0.04 0.940	**0.99** 0.000		**0.80** 0.018	0.66 0.073	**0.97** 0.000	0.31 0.455	−0.60 0.117	**0.93** 0.001
**LYS**		−0.56 0.251	0.67 0.142	0.60 0.206	0.69 0.127	0.85 0.031	0.81 0.049	0.65 0.159	−0.32 0.543	0.74 0.096	0.77 0.071		0.64 0.090	**0.74** 0.034	**0.72** 0.043	−0.67 0.068	**0.84** 0.009
**MET**		0.67 0.144	−0.65 0.158	−0.48 0.341	−0.38 0.454	−0.57 0.240	−0.19 0.726	−0.50 0.311	0.09 0.859	−0.77 0.075	−0.76 0.079	−0.46 0.356		0.65 0.081	0.53 0.179	−0.43 0.284	0.54 0.172
**PHE**		−0.64 0.174	0.75 0.089	**0.90** 0.014	**0.96** 0.002	**0.91** 0.011	**0.96** 0.003	**0.88** 0.022	−0.18 0.730	**0.81** 0.050	**0.82** 0.047	0.78 0.068	−0.32 0.541		0.23 0.580	−0.60 0.116	**0.91** 0.002
**SER**		0.32 0.531	−0.63 0.181	−0.72 0.104	−0.79 0.059	−0.75 0.086	***−0.93*** 0.006	−0.66 0.158	0.42 0.403	−0.57 0.234	−0.61 0.203	−0.68 0.134	0.05 0.924	***−0.91*** 0.011		−0.11 0.789	0.40 0.329
**TRP**		0.79 0.061	−0.17 0.745	−0.51 0.305	−0.54 0.272	−0.38 0.458	−0.17 0.744	−0.57 0.239	−0.56 0.246	−0.49 0.327	−0.43 0.400	−0.17 0.753	0.39 0.446	−0.36 0.483	−0.02 0.965		−0.63 0.097
**TYR**		−0.06 0.906	**0.87** 0.025	0.79 0.062	0.699 0.126	0.76 0.078	0.74 0.095	0.79 0.059	0.23 0.666	0.70 0.120	0.71 0.115	0.46 0.357	−0.25 0.629	0.63 0.182	−0.59 0.221	−0.06 0.912	

**Table cells-09-00139-t003b:** 

(B) Pregnant Rats																	
	H	OGDHC	ALA	ARG	ASP	GABA	GLU	GLY	HIS	ILE	LEU	LYS	MET	PHE	SER	TRP	TYR
C																	
**OGDHC**			−0.17 0.629	−0.03 0.931	−0.06 0.864	−0.12 0.748	−0.22 0.532	−0.15 0.670	0.03 0.944	0.14 0.708	0.19 0.602	0.24 0.506	**0.63** 0.050	0.08 0.832	0.44 0.201	0.05 0.891	0.13 0.725
**ALA**		−0.35 0.569		**0.76** 0.010	0.28 0.439	0.61 0.062	0.10 0.791	**0.77** 0.009	0.60 0.069	**0.83** 0.003	**0.79** 0.006	0.40 0.255	0.40 0.248	0.60 0.067	**0.67** 0.034	−0.02 0.949	**0.73** 0.016
**ARG**		−0.33 0.582	0.86 0.063		**0.03** 0.926	**0.84** 0.002	−0.42 0.226	**0.94** 0.000	0.30 0.406	**0.77** 0.009	**0.64** 0.048	−0.07 0.838	0.46 0.180	**0.91** 0.000	0.36 0.313	−0.52 0.122	0.37 0.298
**ASP**		−0.04 0.953	0.85 0.066	0.85 0.067		0.35 0.323	**0.75** 0.012	0.22 0.548	**0.65** 0.042	−0.11 0.769	−0.10 0.777	0.16 0.653	−0.22 0.544	−0.05 0.885	0.10 0.787	0.58 0.081	0.25 0.478
**GABA**		−0.55 0.340	**0.88** 0.052	**0.96** 0.010	0.82 0.087		−0.17 0.631	**0.96** 0.000	0.44 0.204	0.53 0.118	0.37 0.291	−0.24 0.510	0.16 0.650	**0.80** 0.005	0.14 0.704	−0.45 0.189	0.22 0.548
**GLU**		0.33 0.588	0.49 0.398	0.21 0.732	0.68 0.205	0.23 0.710		−0.25 0.480	0.45 0.192	−0.36 0.306	−0.28 0.432	0.23 0.522	−0.49 0.152	−0.55 0.100	0.02 0.947	**0.89** 0.000	0.11 0.769
**GLY**		−0.51 0.380	0.78 0.118	**0.96** 0.010	0.80 0.104	**0.98** 0.003	0.15 0.808		0.44 0.203	**0.72** 0.020	0.58 0.081	−0.08 0.818	0.32 0.375	**0.87** 0.001	0.31 0.390	−0.49 0.153	0.38 0.274
**HIS**		0.76 0.136	0.24 0.700	0.27 0.660	0.39 0.519	0.01 0.990	0.34 0.576	0.01 0.989		0.30 0.393	0.34 0.341	0.49 0.150	0.27 0.452	0.38 0.279	0.54 0.104	0.30 0.405	0.59 0.075
**ILE**		−0.44 0.456	**0.93** 0.022	**0.98** 0.003	0.85 0.071	**0.98** 0.003	0.26 0.672	**0.95** 0.015	0.17 0.780		**0.98** 0.000	0.46 0.184	**0.74** 0.015	**0.69** 0.028	**0.77** 0.010	−0.39 0.268	**0.77** 0.009
**LEU**		−0.42 0.479	**0.90** 0.036	**0.99** 0.001	0.84 0.076	**0.98** 0.004	0.22 0.724	**0.96** 0.011	0.20 0.752	**0.99** 0.000		0.62 0.054	**0.78** 0.008	0.57 0.084	**0.86** 0.001	−0.27 0.450	**0.86** 0.001
**LYS**		0.07 0.910	0.80 0.106	0.81 0.095	0.71 0.177	0.66 0.230	0.26 0.670	0.61 0.275	0.70 0.191	0.79 0.112	0.80 0.106		0.60 0.068	−0.05 0.898	**0.81** 0.005	0.33 0.353	**0.87** 0.001
**MET**		−0.15 0.812	0.00 0.994	0.39 0.518	−0.08 0.893	0.24 0.698	−0.74 0.156	0.33 0.585	0.21 0.740	0.28 0.651	0.34 0.578	0.39 0.512		0.54 0.111	**0.78** 0.007	−0.31 0.377	**0.66** 0.039
**PHE**		−0.04 0.951	0.50 0.396	0.84 0.077	0.58 0.310	0.67 0.213	−0.14 0.827	0.74 0.149	0.49 0.402	0.72 0.166	0.77 0.125	0.78 0.123	0.76 0.137		0.36 0.307	***−0.64*** 0.046	0.34 0.340
**SER**		0.36 0.549	0.43 0.466	0.60 0.288	0.45 0.446	0.35 0.565	0.02 0.978	0.36 0.546	0.86 0.065	0.50 0.393	0.53 0.353	**0.89** 0.045	0.61 0.272	0.82 0.086		0.12 0.739	**0.90** 0.000
**TRP**		0.85 0.070	−0.21 0.740	−0.45 0.444	0.01 0.988	−0.55 0.339	0.64 0.242	−0.59 0.295	0.53 0.358	−0.46 0.433	−0.48 0.409	−0.08 0.903	−0.63 0.255	−0.43 0.470	0.02 0.979		0.09 0.815
**TYR**		−0.08 0.900	0.81 0.097	**0.96** 0.008	0.86 0.061	0.85 0.068	0.27 0.663	0.86 0.064	0.52 0.373	**0.91** 0.031	**0.93** 0.022	**0.90** 0.035	0.42 0.482	**0.89** 0.041	0.77 0.129	−0.26 0.669	

Pearson’s correlation coefficient (upper value) and *p*-value of the correlation (lower value) are shown. Statistically significant (*p* < 0.05) positive (light grey) and negative (dark grey) correlations are marked. The lower left and upper right triangles refer to the control (C) rats (*n* = 6 in both cases) and the hypoxia-exposed (H) rats (*n* = 8 in A and *n* = 10 in B).

**Table 4 cells-09-00139-t004:** Analysis of the hypoxia-induced changes in interdependence of the levels of amino acids and OGDHC activity in cerebella of the non-pregnant and pregnant rats.

	Groups	Non-Pregnant Rats								Pregnant Rats							
		∑		$\bar{X}$		+		−		∑		$\bar{X}$		+		−	
Parameter		Control	Hypoxia	Control	Hypoxia	Control	Hypoxia	Control	Hypoxia	Control	Hypoxia	Control	Hypoxia	Control	Hypoxia	Control	Hypoxia
**OGDHC**		7.90	12.26	0.53	0.82	0	0	0	11	5.28	2.68	0.35	0.18	0	1	0	0
**ALA**		10.26	12.86	0.68	0.86	5	12	0	1	9.03	7.73	0.60	0.52	3	6	0	0
**ARG**		11.13	12.75	0.74	0.85	8	11	0	1	10.46	7.42	0.70	0.49	5	7	0	0
**ASP**		11.06	12.13	0.74	0.81	7	12	0	1	8.81	3.91	0.59	0.26	0	2	0	0
**GABA**		11.41	12.84	0.76	0.86	8	11	0	1	9.71	6.4	0.65	0.43	5	3	0	0
**GLU**		10.12	11.78	0.67	0.79	6	11	1	1	4.98	5.29	0.33	0.35	0	2	0	0
**GLY**		11.32	12.58	0.75	0.84	8	11	0	1	9.59	7.48	0.64	0.50	4	5	0	0
**HIS**		2.56	12.79	0.17	0.85	0	11	0	1	5.7	6.12	0.38	0.41	0	1	0	0
**ILE**		11.14	12.74	0.74	0.85	6	11	0	1	10.21	8.56	0.68	0.57	6	8	0	0
**LEU**		11.16	12.8	0.74	0.85	6	11	0	1	10.35	8.23	0.69	0.55	6	6	0	0
**LYS**		9.21	11.52	0.61	0.77	0	12	0	0	9.25	5.65	0.62	0.38	2	2	0	0
**MET**		6.53	9.01	0.44	0.60	0	0	0	0	5.57	7.36	0.37	0.49	0	3	0	0
**PHE**		10.81	12.54	0.72	0.84	7	11	1	1	9.17	7.43	0.61	0.50	1	4	0	1
**SER**		8.65	5.45	0.58	0.36	0	1	2	0	7.57	7.18	0.50	0.48	1	6	0	0
**TRP**		5.61	7.89	0.37	0.53	0	0	0	0	6.19	5.45	0.41	0.36	0	1	0	1
**TYR**		8.33	12.6	0.56	0.84	1	11	0	1	10.29	7.27	0.69	0.48	5	6	0	0
**Sum or Average**		**147.20**	**184.54**	**0.61**	**0.77**	**62**	**136**	**4**	**22**	**132.16** #	**104.16**#	**0.55** #	**0.43** #	**38**	**63** #	**0**	**2** #
***p* value of the difference**		0.004		0.004		0.002		0.04		0.004		0.004		0.005		0.18	

For the OGDHC activity (OGDHC) and each of the amino acids, the sum of its correlation coefficients (absolute values) to other amino acids (Σ), average correlation coefficient ($\bar{X}$), and total number of statistically significant positive (+) and negative (−) correlations are shown. At the bottom, the sum (Σs, positive and negative correlations) or average (Xs) of all the values in the row and *p* values of the differences between the parameters in the control and hypoxia groups, estimated by the Wilcoxon signed rank test, are shown, with the statistically significant differences between the control and hypoxia groups in bold. #-Significant (*p* ≤ 0.05) differences between the parameters of the corresponding non-pregnant and pregnant groups.

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
