# Peer review of "Hypoxic Adaptation of Mitochondrial Metabolism in Rat Cerebellum Decreases in Pregnancy"

_cells, 2020, doi:10.3390/cells9010139_

Round 1
Reviewer 1 Report
The work is simple but well designed and results are clear, please provide the following corrections:
On lane 53 you missed the citation after "cerebellar neuronal cells in culture (formatting citation). You should insert the right citation.
In lane 275, in the title you wrote " hysiological consequences of the hypoxia...", You should correct the word.
Do you have some experimental evidence if the observed hypoxic adaptation of the aminoacid metabolism during pregnancy impact on mitochondrial dynamics ? And what happens to mTor ?
It should be interesting to explore these two points in the future.
Author Response
Reviewer 1
On lane 53 you missed the citation after "cerebellar neuronal cells in culture (formatting citation). You should insert the right citation.
In lane 275, in the title you wrote " hysiological consequences of the hypoxia...", You should correct the word.
Answer:
Thank you, corrected
Reviewer 1
Do you have some experimental evidence if the observed hypoxic adaptation of the aminoacid metabolism during pregnancy impact on mitochondrial dynamics? And what happens to mTor ?
It should be interesting to explore these two points in the future.
Answer:
Thank you for these stimulating remarks, we agree that this is a very promising research field. We did not follow the mitochondrial dynamics in this study, but added discussions of these points in lines 363-372 as follows:
“…hypoxic tolerance is associated with the mTOR-dependent autophagy [43]. Increased autophagy may generate the higher amino acid levels after hypoxia (Table 2) to use them for the substrate-level ADP phosphorylation in the hypoxic brain of non-pregnant rats. Autophagy is also coupled to pro-survival function of mitochondrial fission under energy stress [44–46]. However, the homeostatic and pro-survival functions of mTOR, autophagy and mitochondrial fission are highly conditional, with overactivation of these processes also mediating the brain damage by hypoxia [47], 50, 51].
Different levels of activation of mTOR, autophagy and mitochondrial fission may be required for their pro-survival and death-inducing outcomes in different physiological states. In this regard, no increase in the brain amino acid levels by hypoxic exposure of pregnant rats (Table 2) may be due to a higher damaging potential of the autophagy stimulation in this physiological state.”
Reviewer 2 Report
In the manuscript, authors studied the amino acid metabolism in the cerebellum of rats. They compared non-pregnant and pregnant rats under normal and hypobaric hypoxic conditions. Comparison of non-pregnant and pregnant animals revealed that Glu and Trp were significantly upregulated in the pregnant animals. Comparison of normal and hypoxic treated animals revealed the upregulation of Arg, Glu, Lys, Met, Phe, Ser, and Trp in the non-pregnant hypoxic-treated animals. However, no significant change was observed in the pregnant animals. Further, they performed interdependence analysis of OGDHC activity and 15 amino acids. Numbers of significant correlation between the parameters were found in the analysis, and it was also demonstrated that the profiles are highly different between non-pregnant and pregnant animals.
The study thoroughly characterizes the metabolic profile of amino acids comparing non-pregnant and pregnant animals in normal and hypoxic conditions, which potentially provides a previously uncharacterized role of amino acid metabolism in pregnancy or oxygen environment. However, the underlying mechanism remains unclear which requires further investigation.
Points to be analyzed
1. It is not clear what the biological significance of interdependency in amino acids is. Would these value reflect the level of any active product (eg., GABA, serotonin) or physiological activity of the body under different conditions?
2. What would be the mechanism to induce changes in the level of amino acid between non-pregnant and pregnant animals? Would the levels of pregnancy hormones affect the amino acid metabolism?
3. How would hypoxic condition affect the amino acid metabolism? Would hypoxia-inducible factor (HIF) involved in this regulation?
Author Response
Reviewer 2
Points to be analyzed
1. It is not clear what the biological significance of interdependency in amino acids is. Would these value reflect the level of any active product (eg., GABA, serotonin) or physiological activity of the body under different conditions?
Answer:
The interdependence characterizes specific features of metabolic networks, which indeed may differ according to physiological activity required under different conditions. We refer to the publications which show that the level of correlations between metabolites depends on changes in metabolic networks and may be used to decipher specific features of the network. In particular, same average steady-state concentrations of metabolites may be achieved when the concentrations of enzymes and fluxes through the enzymes differ. Yet the difference would be obvious from the correlation analysis, as shown in our work. We added discussion of this point as follows:
“…findings indicate that the pregnancy-imposed changes in the cerebellar amino acid pool (Table 2) and metabolism (Tables 3, 4) are associated with decreased stress adaptability, in line with other studies showing decreased perception of stress in pregnant females [52–54].
It is worth noting that the response of the metabolic network of pregnant rats to hypoxia, undetectable from the average levels of amino acids (Table 2), is evident from the correlation analysis (Table 4). Hence, average levels of cerebellar amino acids are not as sensitive indicators of metabolic changes, as the cumulative parameters characterizing the interdependence of components of the corresponding metabolic network.
It has been noted previously that correlating metabolites may not only be of diagnostic significance [23], but also help deciphering the yet unknown or poorly characterized synthetic and regulatory pathways [20]. “
We also added on the potential mechanism explaining the observed interdependence:
“increased flux of the amino acids degraded through OGDHC (Glu, Gln, Arg, His, Pro) and of the branched-chain amino acids Val and Ile may generate succinyl-CoA for the substrate level phosphorylation in mitochondria. Generation of ATP at the expense of succinyl-CoA may help overcoming energy deficits due to impaired oxidative phosphorylation upon hypoxia. Increased degradation of amino acids in the TCA cycle whose flux is limited by OGDHC, is in good accordance with the negative correlations between the OGDHC activity and the levels of cerebellar amino acids after hypoxia, in contrast to normal metabolism (Table 3A).”
Reviewer 2
What would be the mechanism to induce changes in the level of amino acid between non-pregnant and pregnant animals? Would the levels of pregnancy hormones affect the amino acid metabolism?
Answer:
It is supposed that changed hormonal status generally affects metabolism in pregnancy. However, to the best of our knowledge, specific mechanisms of metabolic changes in the amino acid metabolism in pregnant females, usually observed in studies of diagnostic markers, are poorly characterized. We added the discussion of this issue as follows:
“Hormonal changes during pregnancy are known to coordinate a broad range of physiological adaptations, from the supply of nutrients and oxygen for the fetus growth in utero to specific patterns of parental behaviour[32, 33]. Changed metabolism of amino acids in the brain of pregnant females may be involved in these adaptations, as many of amino acids are neurotransmitters or their precursors.”
How would hypoxic condition affect the amino acid metabolism? Would hypoxia-inducible factor (HIF) involved in this regulation?
Answer:
HIF rather controls the relationship between glycolysis and oxidative phosphorylation. Amino acid metabolism is more related to mTOR. We added the corresponding discussion as follows:
“…hypoxic tolerance is associated with the mTOR-dependent autophagy [43]. Increased autophagy may generate the higher amino acid levels after hypoxia (Table 2) to use them for the substrate-level ADP phosphorylation in the hypoxic brain of non-pregnant rats. Autophagy is also coupled to pro-survival function of mitochondrial fission under energy stress [44–46].”
Reviewer 3
What is known about the human levels of the amino acids and activity of 2-OGDHC during pregnancy? Is it known if any of these changes are relatable to humans?
Answer:
Unfortunately, none of the characterized parameters in the rats can be directly compared to humans. Only pathological brain samples are normally available from humans. However, in the discussion we provide information on studies pointing to the relevance of our animal model to understanding metabolic features also in humans:
“Our previous studies pointed to certain relationship between the amino acid levels in the blood plasma and brain [23], which may be used for translation of findings on the brain samples in animal models to human studies. For instance, increased content of tryptophan in cerebellum of pregnant rats, shown in our study (Table 1) corresponds to findings in humans, which indicate that in maternal plasma, tryptophan catabolites and related compounds change in pregnancy [38, 39]. This study suggests the neurotransmitters-dependent adaptation of maternal body to the fetus growth, similar to our findings (Table 1). Biomarker significance of the amino acid levels in human plasma and urine has been explored also in other studies [40–42]. Their findings point to potential diagnostic significance of the changes in multivariate metabolic profiles, including amino acids, for prediction of gestational diabetes [42]. Besides, the pregnancy-dependent changes in the content of amino acids and their metabolites in plasma and urine point to variation of these parameters, dependent on the increased amino acid demands for the growing fetus [40, 41].”
Reviewer 3
While this study is very interesting, it is not clearly expressed why this is important to investigate. The last paragraph of the discussion starts to hint at the reasons, but is far too brief of a discussion. This needs to be expanded.
Answer:
We formulate this in the Introduction as follows:
“Many amino acids and/or their derivatives are neurotransmitters. Hence, metabolic perturbations in the brain, affecting the levels of amino acids, often have neurological consequences, and vice versa. …
This work is dedicated to elaboration of systemic markers of the changed metabolism of amino acids, resulting from the brain response to the physiological or pathological challenges. To achieve this goal, we consider the two important features of the amino acid metabolism. First, mitochondrial tricarboxylic acid (TCA) cycle actively participates both in the amino acid degradation and de novo biosynthesis of the amino acid precursors, such as 2-oxoglutarate and oxaloacetate. … Second, specific (patho)physiological settings may strongly contribute to different systemic outcomes of the same treatment, because organization of metabolism under these settings may vary.”
Reviewer 3 Report
What is known about the human levels of the amino acids and activity of 2-OGDHC during pregnancy? Is it known if any of these changes are relatable to humans?
While this study is very interesting, it is not clearly expressed why this is important to investigate. The last paragraph of the discussion starts to hint at the reasons, but is far too brief of a discussion. This needs to be expanded.
Are the levels of the amino acids well characterized in the cerebellum during pregnancy in rats, or are the 8 pregnant rat controls the only values known?
Was it noted how many fetuses were present in each pregnant rat? Would any of the levels potentially be impacted based on the number of pups?
What was the %O2 that is reached for the rats in the chamber? The methods are not well described in that section. It is not clear until the results that the time in the chamber is different per rat, based on when they collapse. What is the range of time in the chamber? Does that time differ significantly between groups? What are the %O2 for each group and is this different? Why is this the approach taken rather than having each animal in the chamber for the same amount of time? Following collapse, are they quickly placed in a normal environment? This needs to be described in much more detail.
Author Response
Reviewer 3
Are the levels of the amino acids well characterized in the cerebellum during pregnancy in rats, or are the 8 pregnant rat controls the only values known?
Answer:
We are not aware of other studies of the cerebellar amino acids in pregnant rats.
Reviewer 3
Was it noted how many fetuses were present in each pregnant rat? Would any of the levels potentially be impacted based on the number of pups?
Answer:
Because there was no significant difference between the pups in the two groups, we did not have an opportunity to assess the issue. The relevant information was introduced to Methods section as follows:
“… the number of pups (10±2) did not significantly differ between the control and hypoxic rats, although independent long-term physiological monitoring indicated that the pregnancy failures occurred more often in the latter vs former rats.”
Reviewer 3
What was the %O2 that is reached for the rats in the chamber? The methods are not well described in that section. It is not clear until the results that the time in the chamber is different per rat, based on when they collapse. What is the range of time in the chamber? Does that time differ significantly between groups? What are the %O2 for each group and is this different? Why is this the approach taken rather than having each animal in the chamber for the same amount of time? Following collapse, are they quickly placed in a normal environment? This needs to be described in much more detail.
Answer:
O2 concentration corresponding to the indicated altitude was 5%. We added the requested details of the model description in the Methods (section 2.2)
Round 2
Reviewer 2 Report
N/A
Reviewer 3 Report
The authors took into consideration the recommendations for revision, and have made appropriate changes. The discussion is much more complete following the revision.